# Changes in Cardiovascular and Renal Biomarkers Associated with SGLT2 Inhibitors Treatment in Patients with Type 2 Diabetes Mellitus

**DOI:** 10.3390/pharmaceutics15112526

**Published:** 2023-10-25

**Authors:** Melpomeni Peppa, Aspasia Manta, Ioanna Mavroeidi, Athina Asimakopoulou, Alexandros Syrigos, Constantinos Nastos, Emmanouil Pikoulis, Anastasios Kollias

**Affiliations:** 1Endocrine Unit, 2nd Propaedeutic Department of Internal Medicine, School of Medicine, Research Institute and Diabetes Center, Attikon University Hospital, National and Kapodistrian University of Athens, 12641 Athens, Greece; aspa.manta@gmail.com (A.M.); joannamavroeidi@gmail.com (I.M.); 23rd Department of Internal Medicine, School of Medicine, Sotiria General Hospital, National and Kapodistrian University of Athens, 11527 Athens, Greece; atinasimak@gmail.com (A.A.); kostassyrigos@gmail.com (A.S.); taskollias@gmail.com (A.K.); 33rd Department of Surgery, School of Medicine, Attikon University Hospital, National and Kapodistrian University of Athens, 12641 Athens, Greece; kosnastos@yahoo.gr (C.N.); mpikoul@med.uoa.gr (E.P.)

**Keywords:** SGLT2 inhibitors, cardiorenal syndrome, cardiovascular biomarkers, renal biomarkers, type 2 diabetes mellitus, inflammation

## Abstract

Type 2 diabetes mellitus is a major health problem worldwide with a steadily increasing prevalence reaching epidemic proportions. The major concern is the increased morbidity and mortality due to diabetic complications. Traditional but also nontraditional risk factors have been proposed to explain the pathogenesis of type 2 diabetes mellitus and its complications. Hyperglycemia has been considered an important risk factor, and the strict glycemic control can have a positive impact on microangiopathy but not macroangiopathy and its related morbidity and mortality. Thus, the therapeutic algorithm has shifted focus from a glucose-centered approach to a strategy that now emphasizes target-organ protection. Sodium-glucose transporter 2 inhibitors is an extremely important class of antidiabetic medications that, in addition to their glucose lowering effect, also exhibit cardio- and renoprotective effects. Various established and novel biomarkers have been described, reflecting kidney and cardiovascular function. In this review, we investigated the changes in established but also novel biomarkers of kidney, heart and vascular function associated with sodium-glucose transporter 2 inhibitors treatment in patients with type 2 diabetes mellitus.

## 1. Introduction

The prevalence of type 2 diabetes mellitus (T2DM) has been rising worldwide, reaching epidemic proportions in both high- and middle-income countries [1]. Morbidity and mortality have also been rising dramatically due to diabetic complications, mainly chronic kidney disease (CKD) and cardiovascular disease (CVD) [2].

Diabetic nephropathy is the leading cause of kidney disease globally [3]. Albuminuria and a reduced glomerular filtration rate (GFR) are the main risk factors for end-stage CKD, CVD and death [4].

CVD, specifically ischemic heart disease, coronary artery disease (CAD), heart failure (HF), stroke and peripheral artery disease, accounts for 50% of deaths in patients with T2DM, a number that rises in the context of CKD [5].

Multiple initiatives have been launched to prevent diabetes, achieve optimal glycemic control, and identify and treat diabetic complications as early as possible. Intensive blood glucose control and the management of body weight, blood pressure and lipid levels are all recommended for T2DM patients with tailored treatment objectives and plans [6]. The fundamental component of T2DM therapy continues to be lifestyle modifications, namely adopting a customized food plan and engaging in regular physical activity. However, a significant number of patients fail to adhere to this, and this is occasionally insufficient to protect the target organs [7]. The current list of antidiabetic medications includes metformin, dipeptidyl-peptidase-4 inhibitors (DPP4i), glucagon-like peptide-1 (GLP1) receptor agonists, insulin, sulfonylureas, thiazolinediones and sodium-glucose transporter inhibitors (SGLT2i).

The target-organ protective strategy has taken the place of the glucose-center approach since rigorous monitoring of glucose has failed to lower the frequency of complications, particularly macroangiopathy [8]. Through a recently proposed algorithm, the European Society of Cardiology (ESC) and the European Association for the Study of Diabetes (EASD) have emphasized the importance of taking into account factors other than glucose levels and HbA1c, such as the protection, early detection and treatment of diabetes’ main target organs, the kidney and the heart [9]. GLP-1 receptor agonists are recommended as the first line of therapy for people with risk factors for CVD or those who already have the disease, based on favorable findings from trials with cardiovascular and renal outcomes [10].

SGLT2i remain the first line of treatment in HF and CKD since they have demonstrated remarkable outcomes in heart and kidney failure independent of diabetes, indicating that they act beyond glucose [11]. SGLT2 inhibitors block the SGLT2 transporters and decrease renal glucose absorption in the early proximal renal tubule, leading to glucosuria and reduced blood glucose levels [12,13]. As already mentioned, based on cardiovascular and renal outcomes research, their application has grown dramatically. The Empagliflozin Cardiovascular Outcome Event Trial in Type 2 Diabetes Mellitus Patients (EMPA-REG OUTCOME) was the first study that revealed that SGLT2i improved cardiovascular outcomes in T2DM patients at high cardiovascular risk [14]. According to the DAPA-CKD study, dapagliflozin lowers all-cause mortality and significant adverse renal and cardiovascular events in both diabetic and non-diabetic CKD patients. Dapagliflozin might be initiated in people with an eGFR as low as 25 mL/min, based on the favorable benefit-to-risk ratio in patients with and without T2DM [15]. Numerous other studies, including the Empagliflozin Outcome Trial in Patients with Chronic Heart Failure and a Reduced Ejection Fraction (EMPEROR-Reduced), have confirmed and expanded these findings [16,17,18,19,20,21,22]. SGLT2i are currently included in the 2019 ACC/AHA first-level preventive recommendations for CVD, and the combination of metformin and dapagliflozin is suggested as the optimal therapy regimen for glucose-lowering medication treatment of T2DM [23].

The cardiovascular benefits of SGLT2i are thought to be caused by a number of processes, such as osmotic diuresis and natriuresis, which lower blood pressure, reduce arterial stiffness and vascular resistance, promote weight loss and reduce uric acid and oxidative stress. Elevated hemoglobin and hematocrit levels may also be involved, and glucagon’s control over heart glucose uptake is important for the beneficial inotropic and anti-arrhythmogenic effects [14,24,25]. Another theory states that SGLT2i increase the effectiveness of myocardial activity by switching the metabolism of fuel from inefficient free fatty acids to more effective ketones [26].

The renal protection mechanisms of SGLT2i are also multifactorial. SGLT2i reduce sodium reabsorption in the proximal tubule, increasing sodium delivery to the macula densa. This restoration of tubular glomerular feedback results in reduced kidney blood flow, decreased glomerular hyperfiltration and lowered intra-glomerular pressure [27]. These effects lead to an acute reduction in albuminuria and eGFR, followed by long-term eGFR stability [28]. SGLT2i also positively impact risk factors for renal impairment that have been already mentioned, including high blood glucose, hypertension, serum uric acid and body weight, suggesting potential nephroprotective effects in diabetes patients [29].

According to the definition, a biomarker is “a characteristic that is objectively measured and evaluated as an indicator of normal biological processes, pathogenic processes, or pharmacologic responses to a therapeutic intervention”. Biomarkers are used to detect subclinical disease before it manifests clinically. A biomarker should be easy to test in a non-invasive manner using a well-defined approach, and the measurement obtained should be capable of identifying those who are at risk [30]. Biomarkers can be either a clinical parameter, a circulating molecule (plasma, serum, or urine) or an imaging characteristic [31].

The non-hypoglycemic effects of SGLT2i have been extensively studied and documented in several publications, both in original research and review articles [32,33,34]. In this comprehensive review, we aimed to focus specifically on cardiovascular and renal biomarkers, as a way to provide quantitative insights into biological processes, disease progression and treatment response, and summarize the most recent knowledge on the changes induced by SGLT2i treatment in T2DM patients.

## 2. Heart Biomarkers and SGLT2i Treatment

Heart-related circulating biomarkers are proteins released into the blood either directly from the heart in reaction to tissue-specific damage, or from other cells as a systemic response to heart issues and can be easily detected [35].

### 2.1. Circulating Heart Biomarkers and SGLT2i Treatment

#### 2.1.1. Natriuretic Peptides

Natriuretic peptides (NPs) are hormones that regulate volume and pressure overload in the cardiovascular system, hence preserving homeostasis [36]. Among the three different types of NPs—atrial, released from cardiac atria, C-type, released from endothelial cells, and brain, released from cardiac ventricles—the last one has been proven useful in many clinical settings [37]. The N-terminal prohormone of brain NP (NT-proBNP) has a strong tendency to predict CVD even in asymptomatic adults without a known related history [38], and higher levels have been associated with elevated CV risk [35].

The impact of SGLT2i therapy on NT-proBNP levels in patients with or without T2DM has been thoroughly investigated. Treatment with empagliflozin significantly reduced NT-proBNP levels in both diabetic and non-diabetic patients with HF with reduced ejection fraction (HFrEF) [39,40]. Treatment with empagliflozin, either as a monotherapy [41] or in combination with insulin [42], was beneficial for patients with T2DM hospitalized for acute HF. Over 26 weeks, empagliflozin was also linked to a significant decrease in NT-proBNP levels in patients who had recently experienced myocardial infarction (MI) [43].

Most studies report that canagliflozin significantly decreased the NT-proBNP levels in patients with T2DM, with either clinically stable HF [44,45], or acute HF after hospitalization [46]. Canagliflozin also delayed the rise in NT-proBNP in older patients with T2DM [47]. Compared to glimepiride, only canagliflozin was found to decrease NT-proBNP levels in a subgroup of patients with left ventricular (LV) diastolic dysfunction, suggesting a different therapeutic effect depending on LV function [48].

According to Myhre et al., regardless of baseline NT-proBNP concentrations in HFrEF or HF with preserved ejection fraction (HFpEF), dapagliflozin improved outcomes, with the highest absolute benefit found in patients with higher NT-proBNP levels [49]. Patients with higher baseline NT-proBNP levels also experienced a decrease in NT-proBNP levels when receiving ipragliflozin compared to conventional treatment for T2DM and HFpEF [50].

Although the majority of studies show a significant effect, some studies found no difference in the effects of empagliflozin or canagliflozin in patients with T2DM and clinically stable HF [51,52,53,54]. Dapagliflozin did not affect NT-proBNP in patients with HFrEF with or without DM [55], whereas luseogliflozin failed to show significant results in comparison to voglibose in patients with T2DM and HFpEF [56]. According to Ueda et al., canagliflozin treatment in elderly patients with T2DM and HFpEF produced significant results after four weeks, but not after 24 weeks [57]. The scarcity of data seems to be due to the different populations studied that have various comorbidities and DM features, such as age, diabetes duration or “silent” complications.

#### 2.1.2. Troponin

Troponin (Tn) is a protein that exists in skeletal and cardiac muscle and has three isoforms, troponin C, troponin I and troponin T. The cardiac-specific isoforms of Tn (cTn-I, cTn-T) are exclusively expressed in cardiac muscle [58]. Tn measurement, especially high-sensitivity cTn (hs-cTn), is being used extensively for the diagnosis of acute coronary syndromes, while it also has a prognostic value for CVD morbidity and mortality [35].

Numerous studies have examined the effect of SGLT2i treatment on hs-cTn levels in patients with or without T2DM in various settings of underlying CVD. Vaduganathan et al. compared the effects of canagliflozin and placebo in diabetic patients with known CVD disease and showed that canagliflozin halted the rise in hs-cTnT over six years, while patients with increased hs-cTnT levels had a considerable advantage in terms of significant CVD events [59]. This was also proved in a study of older patients with T2DM, with no prior CVD history, as canagliflozin also delayed the rise in hs-cTnI for over two years [47].

Regardless of hs-cTnT baseline levels, empagliflozin had positive effects on CVD outcomes [60]. Dapagliflozin showed a significantly lowered risk of worsening heart failure or CVD death and tended to reduce the rate of hs-cTnT augmentation with time, independently of baseline levels [61]. In patients with T2DM and CAD, dapagliflozin was found to significantly decrease hs-cTnT levels for six months, compared to vildagliptin [62].

On the other hand, only some data such as that from Griffin et al. did not find a significant effect of empagliflozin on Tn levels in diabetic patients with stable HF [51].

#### 2.1.3. Fibrosis Biomarkers

Soluble suppressor of tumorigenicity 2 (sST2) is a component of the interleukin-1 (IL-1) receptor family [63]. It is generated by the vascular endothelium, cardiomyocytes and cardiac fibroblasts in reaction to stress or damage [64], and seems to be a prognostic factor for HF hospitalization outcomes. Elevated sST2 levels have been linked to HF hospitalization, as well as higher morbidity and mortality [65,66]. Canagliflozin slowed the increase in sST2 levels in patients with T2DM and CVD over six years, while those that had levels of sST2 > 35 ng/mL had a greater benefit for major CVD events [59]. This action has not been proved in older T2DM patients compared to placebo [47]. Empagliflozin also did not affect sST2 in diabetic patients with CAD [67].

Matrix metalloproteinases (MMPs) are a family of enzymes that play a significant part in cardiac remodeling [68], being involved in the development of HF following acute ischemia and used as a predictor of mortality [69]. The only study that investigated the impact of SGLT2i on MMP-2 in diabetic patients with CAD did not report any significant results [67].

### 2.2. Imaging Heart Biomarkers and SGLT2i Treatment

Cardiac imaging is used to evaluate the anatomy and function of the heart and is crucial in the diagnosis, treatment and monitoring of patients with heart diseases [70]. Lately, there has been growing awareness of the impact of the newest treatments on ventricular reverse remodeling.

#### 2.2.1. Left Ventricular (LV) Function Parameters

Left ventricular ejection fraction (LVEF) is a widely used biomarker for evaluating heart function. HF patients are classified into two categories based on LVEF: HF with reduced systolic contraction and thus decreased ejection fraction (HFrEF), and HF with diastolic dysfunction but preserved EF (HFpEF) [71].

Studies examining the impact of SGLT2i therapy on LVEF have produced conflicting results. The EMMY trial investigated the effect of empagliflozin in the acute setting and found a significant improvement in LVEF in patients who underwent percutaneous coronary intervention (PCI) for acute MI [72]. In a group of diabetic patients with HFrEF, canagliflozin showed a significant improvement in LVEF compared to sitagliptin [54]. Empagliflozin demonstrated a difference in T2DM patients with CAD, but this result was not statistically significant [67]. Lastly, empagliflozin had a significant impact on LVEF in non-diabetic patients with HFrEF [73]. Numerous other studies that looked at the impact of various SGLT2i in LVEF in patients with and without T2DM failed to produce any conclusive findings [40,57,74,75,76,77,78,79,80,81,82,83,84,85].

Left ventricular global longitudinal strain (LV-GLS) is one of the newest imaging cardiac biomarkers that examines longitudinal deformation of the left ventricle and is decreased in patients with hypertension and diabetes, despite normal LVEF [86,87,88]. The relationship between SGLT2i therapy and the potential impact on LV-GLS has only been studied in a small number of studies. Nesti et al., found that empagliflozin produced a rapid and sustained amelioration of LV contractility in T2DM patients without CVD, and subclinical dysfunction (LV-GLS < 16.5%) compared with sitagliptin, while no effect was found in patients with normal LV function [89]. Diabetic patients treated with SGL2i for six months showed a significant improvement in LV-GLS [90]. In T2DM patients with stable HF, dapagliflozin was associated with the improvement in LV-GLS and diastolic function [75], an effect that was not obvious when added to metformin. However, a significant difference in global radial strain was noted, suggesting reduced LV contraction [85]. Other studies failed to show a significant effect of treatment with SGLT2i on LV-GLS in T2DM patients with various CVD backgrounds [40,74,83,84].

Stroke volume, which represents the amount of blood that is pumped out of the left ventricle of the heart with each systolic cardiac contraction, does not appear to be affected by SGLT2i treatment [77,79,81,83,85].

#### 2.2.2. LV Volume Parameters

In individuals with HF, LV volumes are related to LV remodeling and deteriorate over time [91]. The most commonly used LV volume parameters are left ventricular end-diastolic volume (LVEDV) and left ventricular end-systolic volume (LVESV).

Results regarding SGLT2i’ impact on LV volume parameters are not entirely consistent, as an improvement in LVEDV was found with SGLT2i treatment mostly in nondiabetic patients. Empagliflozin significantly decreased LV volumes in stable HFrEF patients [73,82], as well as in acute MI within 72 h post PCI [43].

Regarding LVEDV, there are no data supporting an association with SGLT2i treatment in T2DM patients, regardless of their CVD status [40,54,79,80,81,85]. Only Mason et al. found a positive but not even statistically significant impact of empagliflozin in T2DM patients with CAD [67].

Concerning LVESV, significant outcomes were primarily observed in non-diabetic patients on SGLT2 treatment [43,73,82]. Lee et al. found a significant LVESV decrease in T2DM and prediabetic patients with HFrEF after treatment with empagliflozin for 40 weeks [40]. Other studies did not report any statistically significant effects [54,67,78,79,80,81,85].

#### 2.2.3. Left Ventricular Mass (LVM) Parameters

The excessive growth of LVM is known as LV hypertrophy (LVH), and it has been noted as a significant risk factor for CVD and mortality [92].

Several studies support a significant reduction in LVM in T2DM patients treated with different SGLT2i [50,67,79,80,90]. The majority of these patients had underlying CVD, either LVH [79], HFpEF [50] or CAD [67,80]. Studies in nondiabetic patients with stable HFrEF also showed some encouraging outcomes in terms of LVM reduction after treatment with empagliflozin [73,82]. Several studies, however, found no difference [40,76,78,83,93].

#### 2.2.4. Left Atrial (LA) Parameters

Left atrial volume (LAV) depicts diastolic burden and is a reliable indicator of CVD outcomes [94]. In T2DM patients without a history of CVD, increased LAV was a significant independent and incremental predictor of cardiovascular morbidity and mortality [95]. The few studies that investigated the impact of SGLT2i on LAV failed to produce a statistically significant finding [40,76,81,82,83,84,85,93].

#### 2.2.5. LV Diastolic Function

LV diastolic function is evaluated using LV filling pressure parameters, expressed as the ratio of early mitral inflow velocity to early diastolic velocity in the mitral annulus (E/e′) [96].

Treatment with various SGLT2i, in particular canagliflozin, dapagliflozin and empagliflozin, in patients who underwent PCI in the presence of an acute MI had a significant impact on E/e′ [43,75,97]. However, other studies did not support this kind of association between LV diastolic function and SGLT2i treatment [50,54,83,84].

#### 2.2.6. Right Ventricular (RV) Parameters

Unlike LV parameters and function, one study in T2DM patients with CAD treated with empagliflozin did not report any change in the RV mass index (RVMi), RV end systolic and diastolic volumes (RVESVi, RVEDVi) and RV ejection fraction (RVEF) [98].

#### 2.2.7. Plasma Volume (PV)

PV is an important element of CVD pathophysiology. Estimated PV (ePV) is calculated using several formulas based on anthropometric parameters and hematocrit and has been used as a prognostic factor for CV events and mortality [99].

Multiple studies have demonstrated the significant impact of SGLT2i treatment on ePV. Empagliflozin was found superior in lowering ePV, compared to either placebo or conventional glucose-lowering treatment, in T2DM patients with HF or underlying CVD [41,51], exhibiting a positive correlation with NT-proBNP levels [100]. Regardless of baseline LVEF, the same effect has been noticed for canagliflozin, compared to glimepiride in T2DM patients with HF [45].

A summary of both circulating and imaging biomarkers related to heart function affected by SGLT2i treatment is presented in Figure 1.

## 3. Renal Biomarkers and SGLT2i Treatment

### 3.1. Renal Clinical Biomarkers and SGLT2i Treatment

The renoprotective effect of SGLT2i is largely mediated by osmotic diuresis and natriuresis-induced blood pressure (BP) reduction, local renin-angiotensin-aldosterone system inhibition and a reduction in body mass index (BMI) and arterial stiffness [101,102]. Clinically significant decreases in both systolic and diastolic BP have been observed in several randomized controlled studies utilizing SGLT2i in T2DM patients without any compensatory increase in heart rate [103,104,105]. The EMPA-REG outcome study showed that the hemodynamic effects of empagliflozin reduce BP by a reduction in intraglomerular pressure [106].

The regulation of body weight also plays a role in the renoprotective effect of SGLT2i. Both the EMPA-REG OUTCOME trial and the EMPEROR-Reduced trial demonstrated a statistically significant weight loss when SGLT-2 inhibitors were used. In subgroups identified by glucose-lowering medication at baseline and follow-up, a 5 kg decrease in body weight was observed [105]. Patients in the ALTITUDE study slightly decreased their body weight as well [107]. Treatment with dapagliflozin caused significant weight loss, an increase in ketone bodies, an increase in adiponectin, a tendency for high-sensitivity CRP to decline, an increase in glucagon and no change in immunoreactive insulin (IRI), pointing to the possibility that SGLT2i may enhance adipocyte function in obese T2DM patients [108].

### 3.2. Renal Circulating Biomarkers and SGLT2i Treatment

Several regulators of inflammatory response, including nuclear factor-B (NF-B), interleukin 6 (IL-6), monocyte chemoattractant protein 1 (MCP-1), tumor necrosis factor-a (TNF-a), enzymes such as matrix metalloproteinase 7 (MMP-7) and other molecular structures such as fibronectin-1 have been shown to decrease in response to SGLT2i therapy. Among these, TNF-a is considered essential for CKD patients’ ability to manage inflammation. TNF-a may bind to any of the several TNF-Receptor (TNFR) isoforms, including isoforms 1 and 2. Elevated TNFR-1 or TNFR-2, are signs of renal failure. Treatment with canagliflozin was found to reduce TNFR-1 and TNFR-2 levels [107]. As the sole biomarker unrelated to other risk factors for renal function decrease and whose change has been strongly linked to a decline in eGFR, the effect on TNFR-1 appears to be significant [109].

Biomarkers of tubular injury are proteins released in the circulation by the tubular epithelial cells as a response to injury. Among them, kidney injury molecule-1 (KIM-1) is located on the apical membrane of the proximal tubule. It is released into the tubular lumen and absorbed by the peritubular capillaries and is considered a predictor of kidney failure [110]. Compared to placebo, dapagliflozin decreased urinary KIM-1 excretion by 22.6% [111]. Regardless of baseline renal function, ertugliflozin also demonstrated a sustained reduction in KIM-1 in patients with T2DM and stage 3 CKD [112]. Neutrophil gelatinase-associated lipocalin (NGAL) is another biomarker of both proximal and distal tubule injury. Dapagliflozin treatment did not result in NGAL changes [111], however, elevated NGAL levels in patients hospitalized with an acute illness and acute kidney injury treated with SGLT2i indicated that SGLT2i may enhance medullary damage [113]. Lastly, liver-type fatty acid-binding protein (L-FABP), a marker of proximal tubular function, did not change in response to dapagliflozin treatment [111].

In T2DM patients [113], SGLT2i were found to be more effective than placebo at reducing the IgG-to-IgG4 fractional clearance ratio, a measurement of glomerular charge selectivity. However, dapagliflozin therapy had little to no effect on the IgG-to-albumin clearance ratio, a glomerular size selectivity indicator [111].

Glomerular filtration rate (GFR) remains an ideal marker of kidney function. Due to the time-consuming nature of measuring GFR, it is typically estimated using equations that account for endogenous filtration markers such as serum creatinine (SCr) and cystatin C (CysC) [114].

In a cross-sectional study, patients with T2DM who were treated with SGLT2i for at least 24 weeks were compared to those who had not in terms of their estimated glomerular filtration rate (eGFR), which was calculated using both CysC and the estimated glomerular filtration rate (eGFRcys) based on the former. The group receiving SGLT2i had significantly increased eGFRcr while eGFRcys remained the same. The SGLT2i group had a considerably greater difference between eGFRcr and eGFRcys (eGFRcr-cys) [115,116]. Independent of whether or not they had T2DM, the DAPA CKD trial revealed that individuals with CKD who received dapagliflozin had a significantly lower risk of a sustained decline in estimated GFR of at least 50%, end-stage kidney disease or death from renal or cardiovascular causes [117]. The evaluation of ertugliflozin efficacy and safety cardiovascular outcomes (VERTIS CV) trial in T2DM patients with atherosclerotic CVD showed that ertugliflozin was associated with the preservation of eGFR and reduced urine albumin–creatinine ratio (UACR) [21].

At the start of the dapagliflozin treatment, fractional urea excretion was dramatically decreased, and free water clearance was also significantly decreased after four and 14 days [118]. Dapagliflozin also elevated plasma levels of renin, angiotensin II, urine aldosterone, angiotensinogen and copeptin in an immediate and sustained manner [118].

A minor but substantial decrease in HbA1c has been associated with SGLT2i therapy in CKD patients, but not with a decrease in urine fractional glucose excretion [112,119].

### 3.3. Renal Urinary Biomarkers in Treatment with SGLT2i

The urine albumin/creatinine ratio was dramatically decreased after SGLT2i treatment. Urine 8-hydroxy-2C-deoxyguanosine (8-OHdG), a marker of oxidative stress linked to kidney injury, decreased but urinary N-acetyl-b-d-glucosaminidase (NAG), urinary b2-microglobulin and urinary KIM-1 remained unaltered [119,120]. According to Dekkers et al., decreased intra-glomerular pressure or lessened tubular cell damage may be the causes of the albuminuria-lowering effects of SGLT2i treatment [111]. The albumin-to-creatinine ratio changed noticeably more in patients whose mean arterial pressure was under 92 mmHg [121,122].

Proteomics and metabolomics investigations have recently revealed new information about the discovery of novel biomarkers for the progression of diabetic nephropathy. Proteomics comprises methods for studying the urine proteome, whereas metabolomics identifies and quantifies urinary metabolites. The proteomic classifier CDK273 urine test was employed in the PRIORITY trial, and it was found that patients with T2DM had a 2.5-fold higher incidence of microalbuminuria [123].

A summary of biomarkers related to renal function affected by SGLT2i treatment is presented in Figure 2.

## 4. Vascular Biomarkers and SGLT2i Treatment

### 4.1. Arterial Stiffness and SGLT2i Treatment

Arterial stiffness is one of the first indicators of morphological and functional changes in the arterial wall and it is caused by arteriosclerosis rather than atherosclerosis [30]. It has been identified as a significant predictor of CVD events and all-cause mortality and is used by international guidelines regarding arterial hypertension [124]. Arterial stiffness is primarily measured by pulse wave velocity (PWV) and the cardio-ankle vascular index (CAVI).

#### 4.1.1. Pulse Wave Velocity (PWV)

The gold-standard method for the measurement of arterial stiffness is PWV, primarily carotid-femoral (cfPWV) or brachial-ankle (baPWV). In T2DM patients, PWV significantly decreased after treatment with either dapagliflozin [125,126], canagliflozin [127], tofogliflozin [128] or empagliflozin [129,130].

However, other studies that examined short-term or long-term treatment with SGLT2i as monotherapy or in combination with different non-SGLT2i agents did not find a significant change in PWV [131,132]. In a study by Ramirez et al., canagliflozin or perindopril was added in patients with T2DM and hypertension who were already treated with metformin and amlodipine. Carotid-femoral PWV was found to be similarly affected with either canagliflozin or perindopril [133].

#### 4.1.2. Cardio-Ankle Vascular Index (CAVI)

Cardio-ankle vascular index (CAVI) is another biomarker that reflects arterial stiffness and, unlike PWV, is independent of BP. CAVI is now being used as a predictor of CVD [134].

Sakai et al. found a significant reduction in CAVI in T2DM and HFpEF patients treated with SGLT2i (empagliflozin, tofogliflozin, luseogliflozin), with no difference between the SGLT2i groups [135]. In T2DM patients treated with DPP4i for at least one year, a switch to tofogliflozin for six months resulted in a significant improvement in CAVI [136]. On the other hand, in patients with T2DM with inadequate glycemic control, luseogliflozin failed to alter CAVI significantly after 12 weeks of treatment, despite improvement in glycemic control and several metabolic parameters [132].

It should be pointed out that these studies are heterogeneous, with different methodologies and populations. SGLT2i treatment was initiated in T2DM patients with or without CVD risk and/or hypertension, as monotherapy compared to placebo or in different combinations with other antidiabetic medications leading to different findings and somewhat conflicting results.

### 4.2. Endothelial Function and SGLT2i Treatment

The vascular endothelium controls all aspects of vascular balance in reaction to physical and chemical stimuli, through the production of various vasoactive mediators. Endothelial dysfunction plays a role in the formation and progression of atherosclerosis, as well as the rupture of atherosclerotic plaques [137].

Independent of the existence of diabetes, SGLT2 inhibition has been proven beneficial for enhancing endothelial function, which emphasizes the cardioprotective properties of this class of medication even more [138].

#### 4.2.1. Flow-Mediated Dilation (FMD)

FMD is currently the most widely applied method of assessing endothelial function through ultrasound in the brachial artery, in response to changes in blood flow. Although primarily utilized as a research tool, it is considered a predictive factor for patients with both low and high CVD risk [30].

FMD has been extensively studied as a treatment target of SGLT2i. Most studies note a significant improvement in FMD after treatment with different types of SGLT2i [127,135,139,140,141,142], as well as after shifting to SGLT2i from other oral antidiabetic medications [143]. Most of the studies mentioned above were conducted in patients with T2DM and some form of CVD, either HF, CAD or subclinical atherosclerosis. Shigiyama et al. reported significant improvement after 16 weeks of treatment with dapagliflozin as an add-on therapy on metformin in patients with T2DM and no history of CVD who had inadequate glycemic control (HbA1c > 7) [144]. Solini et al. reported that after only 48 h of treatment in T2DM patients, dapagliflozin significantly increased FMD compared to hydrochlorothiazide [126]. Zainordin et al. reported that treatment with dapagliflozin for 12 weeks did not improve FMD, however, the dapagliflozin group appeared to be stable, whereas the placebo group tended to worsen [145]. In diabetic hypertensive patients, dapagliflozin also failed to show a notable difference in brachial FMD [131].

#### 4.2.2. Endothelial Peripheral Arterial Tonometry (EndoPAT)

Peripheral microvascular endothelial function in the fingertip can be measured using an EndoPAT device. This approach involves pulsatile volume fluctuations at the fingertip and is non-invasive [30]. The reactive hyperemia peripheral arterial tonometry index (RHI) can then be calculated. RHI has been used for research purposes.

Treatment with dapagliflozin for six months managed to significantly improve endothelial function, as assessed by RHI in patients with uncontrolled T2DM, compared to non-SGLT2i treatment [146]. However, in a study of 117 patients with T2DM and established CVD, empagliflozin did not significantly affect RHI, despite improvement in glycemic control [147].

### 4.3. Carotid Ultrasonography—Carotid Intima-Media Thickness (cIMT) and SGLT2i Treatment

Carotid ultrasonography is one of the most prevalent ways to detect atherosclerotic changes, and it is effective in both primary and secondary CVD prevention. Carotid intima-media thickness (cIMT) is calculated via carotid ultrasound using a specific reading software and is linked to a high risk of atherosclerosis [30].

Treatment with SGLT2i has thus far been unable to reduce cIMT. In a randomized clinical trial of T2DM patients without CAD, tofogliflozin did not appear superior to conventional (non-SGLT2i) treatment, although antidiabetic medications in general (both SGLT2i and non-SGLT2i) reduced cIMT in a significant way [148]. Ipragliflozin administration also had no effect on the advancement of cIMT, in comparison to conventional treatment for T2DM without the use of SGLT2i [149].

### 4.4. Ankle-Brachial Index (ABI) and SGLT2i Treatment

In comparison to the arms, the lower extremities have greater systolic blood pressure. This connection is represented by the ankle-brachial index (ABI). A reduction in this ratio is used to detect peripheral artery disease and is related to greater CV risk [30].

Both canagliflozin and luseogliflozin failed to affect ABI in a significant way when used in T2DM patients with stable chronic HF or suboptimal glycemic control, respectively [127,132].

### 4.5. Central Hemodynamics/Wave Reflections and SGLT2i Treatment

Central hemodynamic indices are either measurements that estimate central BP parameters and derivatives (central systolic BP, pulse pressure, enhanced pressure and amplification), or parameters that evaluate wave reflections (augmentation index (AIx), forward and backward wave, wave intensity analysis). Wave reflection measurements have regularly been found to be reliable indicators of CVD events and death in high-risk individuals [30].

Studies that examined the effect of treatment with SGLT2i on central BP and the AIx give contradictory results. Both empagliflozin and dapagliflozin reduced central BP and the AIx when compared to placebo for six and 12 weeks, respectively [125,129]. In T2DM patients with high CVD risk, the combination of liraglutide with empagliflozin resulted in a significant decrease in Aix, as well as central BP [130]. Treatment with dapagliflozin for 48 h did not affect either central BP or the augmentation index when compared to hydrochlorothiazide in T2DM patients [126]. When used in hypertensive T2DM patients, dapagliflozin showed a tendency to lower central BP [131]. Concerning pulse pressure, treatment with both empagliflozin and canagliflozin has been found to have a positive effect in T2DM patients with or without hypertension [150,151].

### 4.6. Plasma Biomarkers Related to Vascular Inflammation and SGLT2i Treatment

Inflammation is a major factor in the development of vascular dysfunction in T2DM, which is a key contributor to the high cardiovascular risk associated with the disease. Chronic low-grade inflammation plays an essential part in the malfunctioning of vascular cells that underlies the onset and development of atherosclerosis in diabetes [152]. SGLT2i have demonstrated significant anti-inflammatory properties in addition to their well-known glucose-lowering actions. These medications exhibit a capacity to reduce levels of inflammatory markers and cytokines, contributing to a less inflammatory vascular environment [153].

Widely known as a biomarker of systemic inflammation, high-sensitivity C-reactive protein (hsCRP) has also been associated with the progression of atherosclerotic plaques, especially the more fragile ones [154,155]. Concerning the effect of treatment with SGLT2i on hsCRP, both empagliflozin and canagliflozin showed favorable results in patients with T2DM and CVD [127,140]. Zainordin et al. noted that dapagliflozin, when compared to placebo, actually increased hsCRP, possibly due to the fast decrease in glucose levels observed in that group [145]. However, Sposito et al. showed no difference in hsCRP after treatment with dapagliflozin [141].

Soluble intercellular adhesion molecule-1 (ICAM-1) is a component that has low expression in the endothelium under normal circumstances but is increased during inflammatory processes and may serve as an indicator of early atherosclerosis. Treatment with dapagliflozin gives conflicting results, as Zainordin et al. observed a significant improvement in ICAM-1 in diabetic patients, whereas Sposito et al. note no difference [141,145].

A summary of biomarkers related to vascular function affected by SGLT2i treatment is presented in Figure 3.

## 5. Conclusions

SGLT2i constitute an important class of antidiabetic medications that also exhibit significant cardioprotective and renoprotective effects. Various biomarkers of renal, heart and vascular function, either conventional or novel, which are used to identify subclinical illness before clinical manifestation, are affected by SGLT2i treatment. Thus, it appears that a new area of research has been opened, aiming to use these drugs in subclinical disease to prevent the disease’s evolution or in an attempt to reverse any pathological changes that have already occurred before the disease onset.

## Figures and Tables

**Figure 1 pharmaceutics-15-02526-f001:**
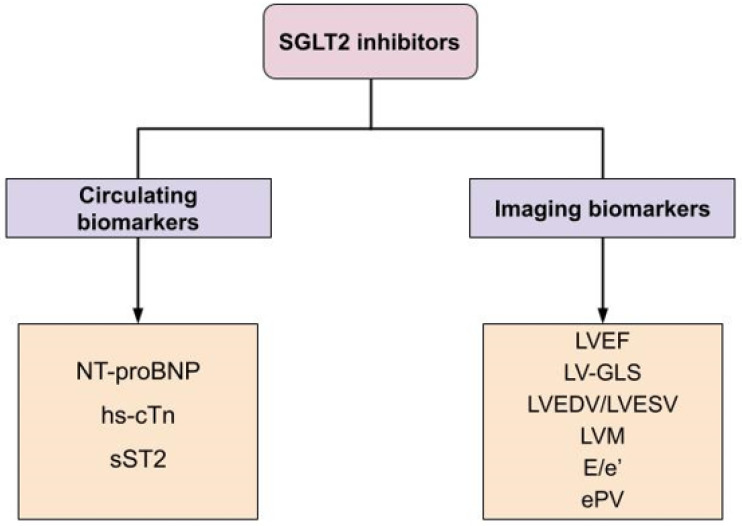
Heart biomarkers affected by SGLT2i treatment. Abbreviations: NT-proBNP: N-terminal prohormone of brain natriuretic peptide; hs-cTn: high-sensitivity cardiac troponin; sST2: soluble suppressor of tumorigenicity 2; LVEF: left ventricular ejection fraction; LV-GLS: left ventricular global longitudinal strain; LVEDV: left ventricular end-diastolic volume; LVESV: left ventricular end-systolic volume; LVM: left ventricular mass; E/e’: ratio of early mitral inflow velocity to early diastolic velocity; ePV: estimated plasma volume.

**Figure 2 pharmaceutics-15-02526-f002:**
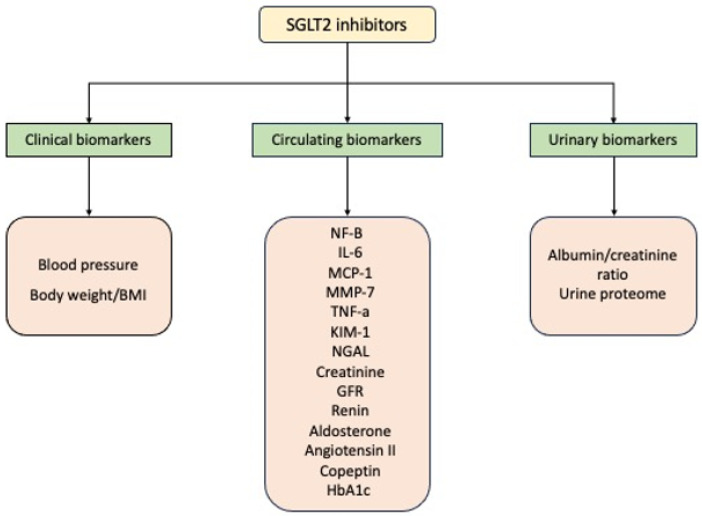
Renal biomarkers affected by SGLT2i treatment. Abbreviations: NF-B: nuclear factor-B; IL-6: interleukin 6; MCP-1: monocyte chemoattractant protein 1; MMP-7: matrix metalloproteinase-7; TNF-a: tumor necrosis factor-a; KIM-1: kidney injury molecule-1; NGAL: neutrophil gelatinase-associated lipocalin; GFR: glomerular filtration rate.

**Figure 3 pharmaceutics-15-02526-f003:**
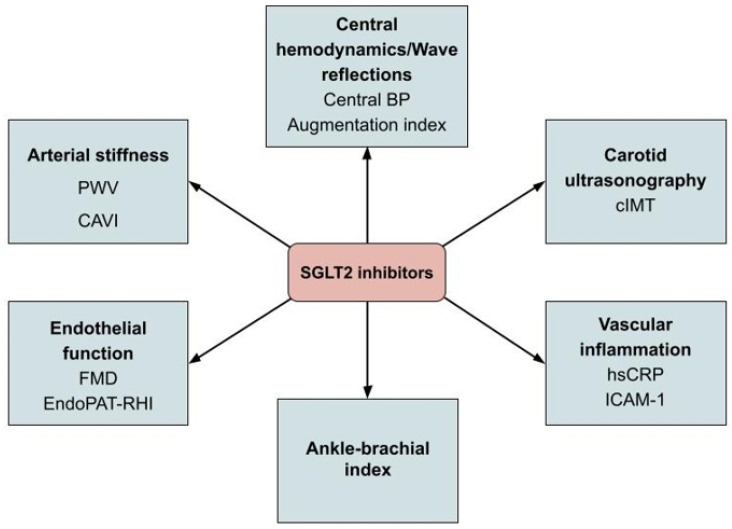
Vascular biomarkers affected by SGLT2i treatment. Abbreviations: PWV: pulse wave velocity; CAVI: cardio-ankle vascular index; FMD: flow-mediated dilation; EndoPAT-RHI: endothelial peripheral arterial tonometry-reactive hyperemia index; BP: blood pressure; cIMT: carotid intima-media thickness; hsCRP: high-sensitivity C-reactive protein; ICAM-1: intercellular adhesion molecule-1.

## Data Availability

Data supporting this article are included within the reference list.

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
