# Peer review of "Changes in Cardiovascular and Renal Biomarkers Associated with SGLT2 Inhibitors Treatment in Patients with Type 2 Diabetes Mellitus"

_pharmaceutics, 2023, doi:10.3390/pharmaceutics15112526_

Round 1
Reviewer 1 Report
This manuscript reviews the non-hypoglycemic effects of SGLT2 inhibitors on biomarkers associated with both the cardiovascular and renal systems
Biomarkers for both systems are described such as natriuretic peptides, troponin etc but whilst their presence with respect to SGLT2 inhibitors is well described, there are no mechanistic insights on how SGLT2 inhibitors are affecting the biomarkers either directly or indirectly, and therefore their relative relevance. There is no interpretation of the results of these observations.
In the cardiovascular system it is divided into those that would be considered true biomarkers natriuretic peptides, troponin etc, and then “imaging heart biomarkers” that I would argue are not biomarkers in their inherent definition “a naturally occurring molecule gene, or characteristic by which a particular pathological or physiological process, disease, etc. can be identified”. This subsequent section is really on functional heart parameters, and whilst important is not in the scope of the title of this article you need to justify why this has been included
The nephrology section is better mechanistically but the differing molecules such as NF-B are part of differing pathways that could be defined such as those in which NF-B stimulation modulates the inflammatory response and TNF-alpha and IL-6 expression. Like my comments above, whilst the changes in FMD and EndoPAT are important in the effects that SGLT2i’s may have, they are not biomarkers of SGLT2i action but rather the hemodynamic changes.
A minor comments
“Type 2 diabetes mellitus seems to be a major health problem” – it is a major health problem
“The major concern seems to be the increased morbidity and mortality due to diabetic complications.” – it is the major concern not seems to be
Please get the manuscript checked by a native English speaker
minon/moderate changes in English grammar is needed
Reviewer 2 Report
This is a well-written manuscript that provides insights into the cardiorenal biomarkers that are the target of SGLT2 inhibitors. Although there are numerous reviews on the subject, this manuscript appears to have approached the subject not from the point of view of the effects of the SGLT2i but rather the targeted biomarkers were the focus. There is still the need to highlight some of those past reviews and indicate how the present review advances the field beyond what is reported in the past reviews.
The authors should check these and cite some of them. Highlight the need for an updated review in this area. See:
1. https://doi.org/10.1016/j.metabol.2021.154918
2. https://doi.org/10.1161/CIRCULATIONAHA.123.065251
3. https://doi.org/10.1111/dom.13648
4. https://doi.org/10.1146/annurev-physiol-031620-095920
It is generally well-written but there is the need to proofread for clarity. Some sections are too verbose.
Reviewer 3 Report
Changes in cardiovascular and renal biomarkers associated with SGLT2 inhibitors treatment, in patients with Type 2 diabetes mellitus is a comprehensive review about the effects of SGLT2i treatment on cardiac and renal markers. It is a valuable contribution to the current medical literature, however, several issues could be improved. My comments are as follows:
1- I think comma is not needed in the title.
2- Abstract is well written and keywords are relevant. However, inflammation could be added as another keyword.
3- Introduction provided adequate background data about the topic and sections of the review are mostly well written. An exception is that authors did not emphasized the role of inflammation in those conditions and the association between SGLT2 inhibitors and inflammation. SGLT2i are drugs of choice in diabetic patients with cardiac conditions, especially heart failure (Expert Review of Endocrinology & Metabolism 18 (3), 255-265). They are also useful in patients with chronic kidney disease (Current Diabetes Reports. 2022; 22(1):39-52). Both, chronic kidney disease (International Journal of Molecular Sciences. 2022; 23(10):5354) and heart conditions (Current Diabetes Reports. 2022;22(1):27-37) are characterized with increased burden of inflammation. On the other hand, it is well established that SGLT2i treatment reduce the burden of inflammation in diabetic subjects (Int J Endocrinol (Ukraine). 2022;18(2):86-9). Emphasize the inflammation in the review, please.
4- Instead of DM2, please use T2DM as an abbreviation of type 2 diabetes mellitus,
5- Conclusions are justified, however, a bit long. If, possible, I recommend summarizing of this section in two or three sentences.
Round 2
Reviewer 1 Report
my queries have been addressed
Reviewer 2 Report
None